# Self-, other-, and meta-perceptions of personality: Relations with burnout symptoms and eudaimonic workplace well-being

Anita de Vries[1¤], Vera M. A. Broks[2], Wim Bloemers[1]*, Jeroen Kuntze[1], Reinout E. de Vries[3]

1 Faculty of Psychology, Open University, Heerlen, The Netherlands, 2 Erasmus MC, University Medical Center, Rotterdam, The Netherlands, 3 Department of Experimental and Applied Psychology, Vrije Universiteit Amsterdam, Amsterdam, The Netherlands

¤ Current address: ARQ IVP, ARQ National Psychotrauma Centre, Diemen, The Netherlands
* wim.bloemers@ou.nl

**Data Availability Statement:** Our data can be accessed through the data steward of the Open University of the Netherlands, Dr. Mellanie Geijsen, email: datasteward@ou.nl. For any further

## Abstract

The present study examined whether disagreement between self-, other-, and meta-perceptions of personality was related to burnout symptoms and eudaimonic workplace well-being. We expected disagreement in personality perceptions to explain incremental variance in burnout symptoms and eudaimonic workplace well-being beyond the main effects of the different personality ratings. Participants were 459 Dutch employees and their 906 colleagues (who provided other ratings of personality). The results, based on polynomial regression with response surface analyses, highlighted strong main effects of self-rated personality traits in relation to burnout symptoms and eudaimonic workplace well-being. This study provides, as far as we know, the first empirical evidence that self-rated Honesty-Humility negatively predicts burnout symptoms. Results showed little evidence on incremental effects of disagreement between personality perceptions, with one clear exception: when respondents misjudged how their colleagues would rate them on Honesty-Humility (i.e., discrepancy between meta- and other-perceptions), respondents experienced more feelings of burnout and less eudaimonic workplace well-being. Our study contributes to the literature by providing evidence that discrepancies between meta- and other-perceptions of Honesty-Humility affect employee well-being (i.e., burnout symptoms and eudaimonic workplace well-being).

## Introduction

Imagine two colleagues at work, Jack and Alice. Jack thinks of himself as somebody who is highly introvert (self-perception). In turn, Jack may believe that Alice thinks of him as relatively neutral with respect to extraversion (meta-perception), whereas Alice actually thinks of Jack as somewhat introverted (other-perception). We wondered if these different perceptions of Jack's personality will affect his well-being at work. Therefore, the present study examined whether discrepancies between self-, other-, and meta-perceptions of personality were related to burnout symptoms and to eudaimonic workplace well-being.

information you can contact the Ethics committee: ceto@ou.nl. There are no ethical or legal restrictions on sharing a de-identified data set. We uploaded our anonymized data set as a Supporting information file.

**Funding:** NO - The funders had no role in study design, data collection and analysis, decision to publish, or preparation of the manuscript.

**Competing interests:** The authors have declared that no competing interests exist.

The question whether (dis)agreement between self- and other-perceptions is related to *individual well-being* has already received significant attention in the past decades [e.g., 1–3]. In general, people prefer to be known and understood by others according to how they see themselves [3, 4]. This desire for coherence suggests that agreement between self- and other-perceptions has positive consequences for individual well-being [1, 5]. Despite a growing interest in the meaning and effects of (dis)agreement between personality perceptions [e.g., 6–8], previous work has focused almost exclusively on self- and other-perceptions. Very little is known about meta-perceptions in this context.

Meta-perception refers to an individual's judgement of how his/her personality is perceived by others [9, 10]. Although it has been previously concluded [e.g., 11] that meta-perceptions are strongly influenced by self-perceptions, empirical findings now suggest that people have some degree of understanding of how others view them that is distinct from their self-perceptions [12, 13]. Still, studies that not only examine self- and other-perceptions, but also include meta-perceptions, remain uncommon [14].

The first aim of our research was to address this gap by including *meta-perception* in addition to self- and other-perceptions. Above self-other disagreement (a), this allowed us to add disagreement between *meta- and other*-perceptions (b), indicating that someone has an *inaccurate* perception of how he or she comes across, and disagreement between *self- and meta*-perceptions (c), indicating that someone thinks that others view him or her in *another* light than the person views him- or herself or indicating that a person is aware that he or she *acts out of character*.

The second aim of our study was to expand the field of interest: no studies have yet been conducted to test possible effects of disagreement in self-, other-, and meta-perceptions of personality in a work environment, relating to *employee* well-being. Therefore, we examined whether disagreement in personality perceptions was related to burnout symptoms (an indicator of poor employee well-being; [15, 16]) and to well-being at work from a eudaimonic perspective. With respect to the latter, eudaimonic well-being at work is recently defined as feeling a sense of meaning and purpose towards work (intrapersonal well-being) and experiencing positive social interactions at work (interpersonal well-being; [17]).

## Agreement and disagreement between self- and other-perceptions

The majority of personality research is still based on self-reports (i.e., self-perceptions: how someone judges his or her own personality). More recently, research has shown that not only self-perceptions, but also other-perceptions of personality (i.e., how someone's personality is judged by others) provide unique information about the role of a person's personality in many domains of psychology [18–21]. Therefore, researchers are increasingly incorporating informant reports into their personality research design. The extent to which other-perceptions do correspond to how people see themselves (self-other agreement; [22]) tends to be moderate for most personality traits, ranging from .30 to .55 [e.g., 18, 23, 24].

The apparent discrepancy between self- and other reports has raised questions about the accuracy of these scores. To date, there is an ongoing discussion about how these discrepancies between self- and other-perceptions can be best understood. For example, discrepancies between self-reports and informant reports are used to discuss which one is more valid [e.g., 18] or as unwanted measurement error to be understood [e.g., 25]. Yet, the present study is based on the assumption that–regardless of the reason–discrepancies in personality perceptions are of psychological relevance [e.g., 6–8].

In particular, the question whether (dis)agreement between self- and other-perceptions of personality is related to individual well-being has received significant attention in the past

decades [e.g., 1–3]. According to the theory of self-verification [3, 4], people have a desire for coherence and therefore prefer to be known and understood by others according to how they see themselves. That is, people prefer self-confirming evaluations, even if the self-view in question is negative [3]. This desire for coherence suggests that agreement between self- and other-perceptions has positive consequences for individual well-being [5]. In addition, research has found that self–other agreement is related to various positive outcomes including better mental health, well-being, and stress regulation (for a review, see [1]).

However, this approach of investigating the effects of self-other (dis)agreement on well-being neglects the role of how someone thinks he or she is viewed by others–and the extent to which this corresponds to other-perceptions–in interpersonal functioning. Therefore, besides self- and other-perceptions, we need to include someone's beliefs about how other people perceive him or her, called meta-perception [9–10].

## Meta-perceptions in addition to self- and other-perceptions

The concept of meta-perception refers to a person's ability to consider another person's impressions [10] and in the case of personality, involves the ability to see one's self from the perspective of another [26]. Meta-perceptions are assumed to be based on mind-reading and perspective-taking processes [9, 22, 11]. For example, Carlson and Kenny [9] suggest that the process of generating meta-perceptions can be described by three stages. First, individuals have to think about who they are, referring to their self-perceptions. Second, individuals need to think about how they behave, based on self-observation information. Third, individuals have to analyze how others do respond to them, integrating information from social feedback processes. Accurate meta-perceptions may help individuals gain self-knowledge and make behavioral changes in response to social cues [9].

Although it has been previously concluded [e.g., 11] that people generally assume that others see them as they see themselves, empirical findings now suggest that people have some insight into how others view them that is distinct from their self-perceptions [12, 13], [see also 27, 28]. Insight tends to be stronger for observable traits (e.g., extraversion) than for less observable (e.g., neuroticism). Nevertheless, people generally understand how others experience them [9, 11]. Given this fact, we believe that it is important to include meta-perception—in addition to self- and other-perceptions—when investigating the meaning and effects of disagreement in personality perceptions. In the following, we focus on disagreement between a) meta- and other-perceptions and b) self- and meta-perceptions, and we outline possible effects of disagreement between these perceptions.

## Possible effects of disagreement between meta- and other-perceptions

The degree to which meta-perceptions are correct (i.e., meta-other agreement), is called meta-accuracy [9]. Meta-accuracy often reflects self-knowledge of one's own personality and behavior, and of how one's behavior differs across social contexts [12, 29, 21]. There is a growing body of recent literature [e.g., 30, 31, 14] examining whether people who score higher on clinical psychopathological measures have different meta-perceptions than people from non-clinical groups. Such research showed that people with personality disorders tend to be less accurate and tend to overestimate the negativity of the impressions they make on others [31, 14]. In turn, relative to people who are less psychologically adjusted, people who are more adjusted tend to have greater meta-accuracy [30].

Little is known of the *effects* of disagreement between meta- and other-perceptions of personality (for an exception, see [28]). When meta- and other-perceptions of personality traits do not overlap, this may indicate that someone has an *inaccurate* perception of how he or she

comes across. In other words, meta- and other discrepancies may refer to having a false understanding of how someone is viewed by others. Where accurate meta-accuracy reflects knowledge of one's own social identity or reputation [29], people with inaccurate meta-accuracy might not notice important differences in their reputation across situations. Someone may be unduly optimistic about his or her reputation, whereas others in fact are far more critical. Alternatively, when someone has a tendency to downgrade oneself, others may be far more positive about him or her. In either case, discrepancies between meta- and other-perceptions may make people feel misunderstood and may have important implications for interpersonal functioning [30], because most people adjust their behavior to how they think others perceive them [9]. For example, people who think that others view them negatively are more likely to start interactions with them in a hostile manner [32].

Because of this importance of accurate meta-perceptions for interpersonal functioning, one would expect that they will also play an important role in work settings. However, surprisingly, research regarding meta-perceptions in work settings is very scarce (for an exception, see [33]). Within a work environment, discrepancies between meta- and other-perceptions of personality may indicate that such employees have a false understanding of how they are viewed by colleagues. These employees may have less self-knowledge and may have a difficult time responding to social cues at work [9]. This may result in stress and less inter- and intrapersonal well-being. Therefore, we expected that disagreement between meta- and other-perceptions is related to *more* burnout symptoms and *less* eudaimonic workplace well-being.

## Possible effects of disagreement between self- and meta-perceptions

It is yet unknown what, if any, effects are to be expected in the case of disagreement between self- and meta-perceptions of personality. Discrepancy between self- and meta-perceptions may indicate that someone thinks that others view him or her in *another* light than the person views him- or herself or that a person is aware that he or she *acts out of character*. According to Carlson, Vazire, and Furr [13], thinking that others view one in another light than one views oneself, may have negative consequences for mental health. For example, self-perceptions of personality may be more positive than one's meta-perceptions (e.g., narcissists might believe that others do not recognize how great they are; [34]), or in other cases, meta-perceptions might be more positive than self-perceptions (e.g., people suffering from low self-esteem). Within the work context, previous studies have shown that self-esteem—defined as the confidence in one's own worth or abilities—is related to more effective work relationships and contributes to a pleasant work environment [35], whereas poor self-esteem leads to more mental health issues, such as burnout [e.g., 36].

From another point of view, disagreement between self- and meta-perceptions may indicate that someone is aware that he or she *acts out of character*. Within a work environment, possible negative effects of acting out of character are emphasized in research on emotional labor [37], [see also 38]. Morris and Feldman [39] defined emotional labor as the "effort, planning, and control needed to express organizationally desired emotion during interpersonal transactions" (p. 987). Several studies suggest that emotional labor, for example when employees have to suppress their felt emotions (i.e., surface acting; [37]), can be both exhausting and stressful, and decreases psychological well-being (for a review, see [40]). The results of a recent review indicate that emotional labor is a job stressor that leads to burnout [41].

In the present study, we assumed that acting out of character at work—similar to emotional labor—is exhausting and may decrease well-being. Based on this assumption and other above-mentioned research, we expected that discrepancies between self- and meta-perceptions of personality are related to *more* burnout symptoms and *less* eudaimonic workplace well-being.

## The present study

The main purpose of this study was to explore whether disagreement between self-, other-, and meta-perceptions of personality was positively related to burnout symptoms and negatively related to eudaimonic workplace well-being. We measured how employees judged their own personality (self-perception), how employees were judged by their colleagues (other-perception), and how employees thought they were judged by their colleagues (meta-perception). This allowed us to examine possible effects of disagreement between a) self- and other-perceptions, b) meta- and other-perceptions, and c) self- and meta-perceptions in relation to burnout symptoms and eudaimonic workplace well-being.

The most widely used framework for personality research is the Big Five model [42] or Five Factor-Model [43]. However, in the last two decades, a new personality model has been introduced: the *HEXACO personality model* [44–46]. The most important feature of the HEXACO model is the addition of a sixth dimension of personality: Honesty-Humility. This dimension reflects individual differences in tendencies to be sincere, fair, and unassuming versus manipulative, greedy, and pretentious [47]. Interestingly, Honesty-Humility has emerged as an important trait in understanding a number of important criteria, such as cooperation and prosocial behavior [48], moral disengagement [49], and counterproductive work behavior [e.g., 50, 51]. In view of the fact that research has shown that the HEXACO framework—with its separate factor for Honesty-Humility—provides a more encompassing empirical and theoretical account of personality variation than the Big Five framework [45], we decided to conduct our study within the HEXACO framework.

Past research on (dis)agreement in personality perceptions typically relied on methods that used profile similarity indices or absolute difference scores. However, these commonly used methods may produce incomplete and/or inaccurate results and suffer from numerous methodological problems [see 52]. Therefore, we used *polynomial regression analysis* with response surface analysis [53], [see also 54]. This statistical approach allowed us to achieve more articulated answers to the question whether disagreement in personality perceptions was related to burnout symptoms and eudaimonic workplace well-being, and allowed us to examine whether disagreement in personality perceptions explained additional variance beyond the main effects of self-, other-, and meta-personality perceptions. This latter was particularly relevant for our study given the large amount of evidence from meta-analyses that self-perceptions of personality traits are robust predictors of burnout [36, 55] and well-being [56].

## Methods

### Procedure

In the period from November 2019 to March 2020, 15 student members of a research team at the Open University in the Netherlands collected data for the current study. To obtain a diverse sample, the students approached potential participants in their own networks (family, friends, and colleagues). Information letters were used for recruitment, stating that participants (minimum age: 18 years) were required to be presently employed and to have at least three colleagues with whom they often collaborated. Potential participants were asked to complete (personality) questionnaires at two different moments and to obtain personality ratings from three close colleagues. The information letter provided detailed information about the study and emphasized that participation was voluntary, confidential, and could be stopped at any time. In addition, it was clearly stated that participants did not have access to the answers of their colleagues, and vice versa. The information letter contained a link to the first online questionnaire (T1) and started with an informed consent question, followed by background

variables, a self-perception personality inventory, and questions measuring work-related burn-out symptoms and eudaimonic workplace well-being. Next, participants received a second information letter, asking them to approach three colleagues for informant reports on the personality inventory (T2). This second letter for colleagues also provided detailed information about the study and a link to the online questionnaire, which started with an informed consent question. One week after having finished the T1-questionnaire, participants received an e-mail with access to the meta-perception personality inventory (T3). In exchange for participation, respondents were offered individualized personality reports (based on their self-perception). The entire procedure of data collection was approved by the Ethical Committee of the Open University of the Netherlands (correspondence November 5, 2019, registration number: U/2019/08651).

## Participants

In total, 435 participants and 963 of their colleagues (who provided informant reports) partici-pated in this study. After three data quality checks (see S1 Text and S1 Table) and subsequent exclusion of respondents, the final sample included 359 participants ($M_{age}$ = 40.3, $SD$ = 12.5, 68.8% female), providing each two or three informant reports (906 in total). Of our partici-pants, 21.7% had lower or intermediate vocational education, 45.1% had higher vocational education and 33.2% had a university degree. Participants worked in a variety of sectors, such as social work (18.4%), health care (10.0%), and government (8.1%). The 906 informants ($M_{age}$ = 44.3, $SD$ = 12.2, 55.3% female) reported the following educational levels: 24.2% lower or intermediate vocational education, 45.5% higher vocational education and 30.2% university degree. Of the 359 participants, 348 completed the meta-perception personality inventory.

## Measurements

**Personality.** Self- and other-perceptions of personality traits were assessed using the Dutch self-report and observer report versions of the HEXACO Personality Inventory–Revised (HEXACO-PI-R; [57]). The HEXACO-PI-R measures the following six personality dimensions: Honesty-Humility, Emotionality, Extraversion, Agreeableness, Conscientious-ness, and Openness to Experience. Each personality domain scale consists of four facet-level scales. For reasons of efficiency, we used the *short* version of the HEXACO-PI-R [58], which contains 96 statements. Responses were assessed with a 5-point Likert response scale from 1 (*strongly disagree*) to 5 (*strongly agree*). In line with previous research [58], the present study showed that the psychometric properties of the HEXACO-PI-R domain scales were adequate (see S2 Table).

To measure meta-perceptions, we adapted the HEXACO-PI-R to assess how participants estimated their colleagues' perception of them. The inventory consisted of the 96 statements of the self-report version of the HEXACO-PI-R, but were preceded by the phrase: 'My colleagues think I am someone who. . .'. For example: '*My colleagues think I am someone who avoids mak-ing small talk with people*' and '*My colleagues think I am someone who prefers to do whatever comes to mind, rather than stick to a plan*'. Principal components analysis of the 24 facet-level scales yielded six components, clearly identified as the six HEXACO dimensions (see S3 Table). Psychometric properties of these HEXACO-PI-R domain scales were adequate, with Cronbach's alpha reliabilities ranging from .82 for Emotionality and Extraversion to .88 for Agreeableness (see S2 Table).

**Burnout symptoms.** To measure the degree of burnout symptoms, we used the validated Dutch work-related version of the Burnout Assessment Tool (BAT; [15]). The BAT consists of 23 items, divided into four subscales: exhaustion (8 items), mental distance (5 items,)

emotional disorder (5 items), and cognitive disorder (5 items). Previous psychometric research showed that these four subscales are clearly distinguishable, and, in turn, refer to one underlying construct [15]. Therefore, based on the unit-weighted combination of the four subscales, one composite score was made as a general indicator for burnout symptoms. Examples of items are: '*At work, I feel mentally exhausted*' and '*I am cynical about what my work means to others*'. The items were answered on a 5-point Likert scale, ranging from 1 *(never)* to 5 *(always)*. In our sample, the Cronbach's alpha reliabilities of the subscales of the BAT (exhaustion, mental distance, emotional disorder, and cognitive disorder) were respectively .88, .80, .84, and .78, showing adequate Cronbach's alpha reliability. Total Cronbach's alpha reliability over all 23 items was .92 (see S2 Table).

**Eudaimonic workplace well-being.**   We used the validated Eudaimonic Workplace Well-being Scale (EWWS; [17]) to measure workplace well-being from a eudaimonic perspective. The EWWS consists of 8 statements, equally divided over two dimensions. The *intra*personal dimension focuses on an employees' energy, purpose, and personal growth. An example item is: '*I feel that I have a purpose at my work'*. The *inter*personal dimension focuses on the comfort an individual feels at work and on the presence of relationships with others. An example item is: '*I feel connected to others within the work environment'*. Ratings were made on a 5-point Likert scale, ranging from 1 (*strongly disagree*) to 5 (*strongly agree*). The current study yielded adequate Cronbach's alpha reliabilities for the overall scale (.81), the intrapersonal dimension (.74), and the interpersonal dimension (.84) (see S2 Table).

## Data analysis

In the present study, we used polynomial regression analysis with response surface analysis [53], [see also 54]. This statistical approach allowed us to examine the extent to which combinations of two predictor variables (*X, Y*) relate to an outcome variable (*Z*), and allowed us to examine whether disagreement in personality perceptions explained additional variance beyond the main effects of self-, other-, and meta-personality perceptions [53].

We started by assessing the presence of sufficient discrepancies between personality ratings in the data (see S1 Text). S4 Table shows that the percentage of discrepancies between personality ratings in either direction exceeds the required 10% for each personality factor scale [54]. Next, polynomial regression with response surface analyses were conducted [53, 54].

First, we centered the personality factor scales around the *midpoint* for each scale, as recommend for this type of analysis [59, 52], [see also 54, 60]. Therefore, we subtracted 3 from each original (not standardized) personality factor score, because ratings were measured on a 5-point Likert scale.

Second, based on the centered personality factor scores (*X, Y*), we created three new variables for each personality factor across the different ratings: the square of the centered personality factor score from one rating ($X^2$); the square of the centered personality factor score from another rating ($Y^2$); and the cross-product of the two related centered personality factor scores (XY). The full polynomial equation is [53]: $Z = b0 + b1X + b2Y + b3X^2 + b4XY + b5Y^2 + e$. In line with Luan and Bleidorn [2], each type of (dis)agreement for each HEXACO personality factor was modeled separately.

Third, again in line with Luan and Bleidorn [2], for each type of (dis)agreement (i.e., self-other, self-meta and meta-other) and for each HEXACO personality factor, we compared three models using the Akaike Information Criterion (AIC) to avoid overfitting the data. Model 1 controlled for age and gender. Model 2 added the linear effects of two types of personality ratings (i.e., *b1X* and *b2Y*), examining the unique predictive power of the two personality ratings separately. To examine whether (dis)agreement in personality perceptions explained

additional variance beyond the main effects, Model 3 added the polynomial regression coefficients, namely quadratic effects of the two personality ratings (i.e., $b3X^2$ and $b5Y^2$) and an interaction effect between the linear effects of both ratings (i.e., $b4XY$).

Fourth, we conducted response surface analysis when model comparison tests indicated that the full model (i.e., Model 3) fitted the data best. To aid interpretation of the three-dimensional relations, for each significant polynomial regression model, we examined slope and curvature and we plotted the three-dimensional response surface [61]. Therefore, the model coefficients (which are *not* interpretable in isolation when conducting polynomial regression models), were transformed into four surface test values: $a_1$ to $a_4$ [54]. This allowed interpretation of: whether and how the *linear* ($a_1$) and *nonlinear* ($a_2$) relation between the agreement in personality perceptions was related to burnout and well-being; whether and how the *direction* of the discrepancy ($a_3$) between personality perceptions was related to burnout and well-being; and whether and how the *degree* of discrepancy ($a_4$) between personality perceptions was related to burnout and well-being. Note that $a_1$ and $a_2$ concern the line of perfect *agreement* and $a_3$ and $a_4$ concern the line of perfect *incongruence* on the modelled surface. For the current study, we were especially interested in the line of perfect incongruence: the direction and degree of *discrepancy* between personality perceptions.

## Results

Table 1 presents the AIC values of all models. The optimal model is selected based on the lowest AIC-value. For the best-fitting models, Tables 2 and 3 shows the (polynomial) regression coefficients. Note that in cases where the full model (i.e., Model 3) was the optimal model, polynomial regression coefficients are *not* interpretable in isolation, therefore we added the surface test values ($a_1$ to $a_4$).

### Self-, other-, and meta-perceptions of personality in relation to burnout symptoms

In relation to burnout symptoms, Model 2 was the best-fitting model in most cases. The results in Table 2 indicated strong main effects of *self-rated* personality traits in relation to burnout. Specifically, self-perceptions of Honesty-Humility (β = -.29, $p < .001$), Extraversion (β = -.32, $p < .001$), Agreeableness (β = -.23, $p < .001$), and Conscientiousness (β = -.27, $p < .001$) were significantly and negatively related to burnout. In addition, self-perceptions of Emotionality significantly and *positively* predicted burnout (β = .28, $p < .001$). Other-perceptions of personality were *not* significantly related to burnout. Although the results seemed to indicate that meta-perceptions did have predictive power (in de case of meta- and other-perceptions), when self- and meta-perceptions were taken together, it became clear that especially self-ratings explained a significant amount of variance in the prediction of burnout, and that meta-perceptions did not predicted above and beyond the variance explained by self-perceptions.

In four cases, Model 3 was the best-fitting model, indicating that (dis)agreement in personality perceptions explained additional variance beyond the main effects. First, the results in Table 2 revealed that the *degree* ($a_4 = 0.61$, $p = .039$) of the *discrepancy* between meta- and other-perceptions of Honesty-Humility was associated with burnout. The positive direction coefficient indicates that burnout increased more sharply as the degree of discrepancy (regardless of the direction) increased (see Fig 1a). Second, with respect to self- and meta-perceptions of Emotionality, the results indicated that when *in agreement*, self- and meta-perceptions of Emotionality were positively related to burnout ($a_1 = 0.35$, $p < .001$; see Fig 1b). Third, the results also showed that as self- or meta-perceptions on the one hand and other-perceptions of

**Table 1. Results of model comparison tests: Self-, other-, and meta-perceptions of HEXACO personality traits in relation to burnout symptoms en eudaimonic workplace well-being.**

| | HEXACO factor scales | Perceptions (X—Y) | AIC (Model 1) | AIC (Model 2) | AIC (Model 3) |
|---|---|---|---|---|---|
| **Burnout** | Honesty-Humility | self—other | 450 | **418** | 422 |
| | | meta—other | 431 | 412 | **411** |
| | | self—meta | 431 | **396** | 398 |
| | Emotionality | self—other | 450 | **406** | 410 |
| | | meta—other | 431 | **405** | 410 |
| | | self—meta | 431 | 389 | **387** |
| | Extraversion | self—other | 450 | **415** | 421 |
| | | meta—other | 431 | **416** | 421 |
| | | self—meta | 431 | **397** | 402 |
| | Agreeableness | self—other | 450 | **429** | 433 |
| | | meta—other | 431 | **413** | 416 |
| | | self—meta | 431 | **405** | 408 |
| | Conscientiousness | self—other | 450 | **424** | 428 |
| | | meta—other | 431 | **416** | 420 |
| | | self—meta | 431 | **406** | 408 |
| | Openness to Experience | self—other | 450 | 451 | **440** |
| | | meta—other | 431 | 429 | **422** |
| | | self—meta | 431 | **429** | 433 |
| **Eudaimonic workplace well-being** | Honesty-Humility | self—other | 570 | 561 | **559** |
| | | meta—other | 548 | 536 | **529** |
| | | self—meta | 548 | **534** | 536 |
| | Emotionality | self—other | 570 | **563** | 566 |
| | | meta—other | 548 | **545** | 551 |
| | | self—meta | 548 | 539 | **538** |
| | Extraversion | self—other | 570 | **515** | 518 |
| | | meta—other | 548 | **519** | 520 |
| | | self—meta | 548 | **502** | 506 |
| | Agreeableness | self—other | 570 | **558** | 563 |
| | | meta—other | 548 | **538** | 543 |
| | | self—meta | 548 | **536** | 540 |
| | Conscientiousness | self—other | 570 | 559 | **558** |
| | | meta—other | 548 | **545** | 545 |
| | | self—meta | 548 | **538** | 538 |
| | Openness to Experience | self—other | 570 | **570** | 573 |
| | | meta—other | 548 | **544** | 548 |
| | | self—meta | 548 | **545** | 548 |

*Note*: Best-fitting models (based on AIC values) in bold. Model 1 = control variables (i.e., age and gender). Model 2 = control variables + $X$ and $Y$. Model 3 = control variables + $X$, $Y$, $X^2$, $XY$, and $Y^2$.

Openness to Experience on the other hand that were *in agreement* increased, burnout *curvilinearly* increased ($a_2 = 0.24$, $p < .001$ respectively $a_2 = 0.22$, $p = .005$). Fig 1c and 1d shows that congruent combinations of extreme (versus moderate) levels of Openness to Experiences were related to more burnout.

**Table 2. Self-, other-, and meta-perceptions of HEXACO personality traits in relation to burnout symptoms.**

| HEXACO factor scales | Perceptions $X$—$Y$ | Best-fitting model ($R^2$) | $X$ | $Y$ | $X^2$ | $XY$ | $Y^2$ |
|---|---|---|---|---|---|---|---|
| H | self—other | Model 2 (.13) | **-.29, _p_ < .001** | -.04, _p_ = .498 | | | |
| | | | **[-0.39, -0.19]** | [-0.17, 0.08] | | | |
| | meta—other | Model 3 (.12) | .05, _p_ = .697 | -.18, _p_ = .232 | -.01, _p_ = .880 | -.36, _p_ = .025 | .26, _p_ = .035 |
| | | | | $a_1$ = -0.13, _p_ = .475 | $a_2$ = -0.11, _p_ = .414 | $a_3$ = 0.23, _p_ = .293 | **$a_4$ = 0.61, _p_ = .039** |
| | self—meta | Model 2 (.14) | **-.29, _p_ < .001** | -.05, _p_ = .466 | | | |
| | | | **[-0.42, -0.15]** | [-0.19, 0.09] | | | |
| E | self—other | Model 2 (.16) | **.28, _p_ < .001** | .10, _p_ = .146 | | | |
| | | | **[0.18, 0.38]** | [-.0.04, 0.24] | | | |
| | meta—other | Model 2 (.12) | **.19, _p_ = .002** | .15, _p_ = .071 | | | |
| | | | **[0.07, 0.31]** | [-.01, 0.30] | | | |
| | self—meta | Model 3 (.18) | .20, _p_ = .010 | .14, _p_ = .101 | .03, _p_ = .796 | -.39, _p_ = .092 | .33, _p_ = .018 |
| | | | | **$a_1$ = 0.35, _p_ < .001** | $a_2$ = -0.03, _p_ = .715 | $a_3$ = 0.06, _p_ = .702 | $a_4$ = 0.75, _p_ = .095 |
| X | self—other | Model 2 (.13) | **-.32, _p_ < .001** | .03, _p_ = .629 | | | |
| | | | **[-0.43, -0.21]** | [-0.11, 0.17] | | | |
| | meta—other | Model 2 (.09) | **-.20, _p_ = .002** | -.05, _p_ = .512 | | | |
| | | | **[-0.33, -0.07]** | [-0.20, 0.10] | | | |
| | self—meta | Model 2 (.14) | **-.35, _p_ < .001** | .07, _p_ = .409 | | | |
| | | | **[-0.51, -0.20]** | [-0.10, 0.23] | | | |
| A | self—other | Model 2 (.10) | **-.23, _p_ < .001** | .00, _p_ = .940 | | | |
| | | | **[-0.32, -0.13]** | [-0.11, 0.11] | | | |
| | meta—other | Model 2 (.10) | **-.20, _p_ < .001** | .00, _p_ = .940 | | | |
| | | | **[-0.29, -0.10]** | [-0.11, 0.12] | | | |
| | self—meta | Model 2 (.12) | **-.18, _p_ = .005** | -.07, _p_ = .232 | | | |
| | | | **[-0.31, -0.06]** | [-0.19, 0.05] | | | |
| C | self—other | Model 2 (.11) | **-.27, _p_ < .001** | .00, _p_ = .975 | | | |
| | | | **[-0.38, -0.16]** | [-0.13, 0.12] | | | |
| | meta—other | Model 2 (.09) | **-.20, _p_ = .001** | .00, _p_ = .969 | | | |
| | | | **[-0.31, -0.09]** | [-0.14, 0.14] | | | |
| | self—meta | Model 2 (.11) | **-.26, _p_ = .002** | -.01, _p_ = .918 | | | |
| | | | **[-0.42, -0.10]** | [-0.16, 0.14] | | | |

(_Continued_)

**Table 2.** (Continued)

| HEXACO factor scales | Perceptions $X$—$Y$ | Best-fitting model ($R^2$) | $X$ | $Y$ | $X^2$ | $XY$ | $Y^2$ |
|---|---|---|---|---|---|---|---|
| O | self—other | Model 3 (.09) | .09, $p$ = .079 | -.10, $p$ = .154 | -.12, $p$ = .068 | .21, $p$ = .163 | .16, $p$ = .157 |
| | | | | $a_1$ = -0.01, $p$ = .910 | $a_2$ = **0.24, $p$ < .001** | $a_3$ = 0.19, $p$ = .079 | $a_4$ = -0.17, $p$ = .561 |
| | meta—other | Model 3 (.09) | .08, $p$ = .208 | -.07, $p$ = .319 | -.12, $p$ = .164 | .23, $p$ = .194 | .12, $p$ = .364 |
| | | | | $a_1$ = 0.00, $p$ = .926 | $a_2$ = **0.22, $p$ = .005** | $a_3$ = 0.15, $p$ = .224 | $a_4$ = -0.23, $p$ = .499 |
| | self—meta | Model 2 (.05) | -.04, $p$ = .543 | .14, $p$ = .076 | | | |
| | | | [-0.18, 0.09] | [-0.02, 0.29] | | | |

*Note*: HEXACO factor scales are Honesty-Humility (H), Emotionality (E), Extraversion (X), Agreeableness (A), Conscientiousness (C), and Openness to Experience (O). $R^2$ refers to the variance explained of the model. The table represents the estimates and 95% confidence intervals of unstandardized regression coefficients, with significant effects ($p$ < .05) in bold. Model 1 = control variables (i.e., age and gender). Model 2 = control variables + $X$ and $Y$. Model 3 = control variables + $X$, $Y$, $X^2$, $XY$, and $Y^2$. The model coefficients of Model 3 are not interpretable in isolation, but used to compute $a_1$–$a_4$. Surface test values $a_1$ and $a_2$ represent the slope and curvature of the line of agreement and $a_3$ and $a_4$ represent the slope and curvature of the line of incongruence.

## Self-, other-, and meta-perceptions of personality in relation to eudaimonic workplace well-being

Table 3 reveals that especially Extraversion explained a significant amount of variance in the prediction of workplace well-being, such that self-perceptions of Extraversion (β = .47, $p$ < .001) significantly and positively predicted workplace well-being. Moreover, higher levels of self-rated Agreeableness (β = .24, $p$ < .001) and Conscientiousness (β = .24, $p$ = .006) also predicted more workplace well-being. Other-perceptions generally showed no unique predictive power.

In four cases, Model 3 was the best-fitting model. First, with regard to Honesty-Humilty, the results suggested that disagreement between self- and other-perceptions and between meta- and other-perceptions did show unique predictive power in relation to *workplace well-being*. In both cases, the *degree* ($a_4$ = -0.79, $p$ = .005 respectively $a_4$ = -1.07, $p$ = .002) of the *discrepancy* between perceptions of Honesty-Humility was associated with *less* workplace well-being. These findings indicated that workplace well-being decreased more sharply as the degree of discrepancy (regardless of the direction) increased (see Fig 1e and 1f). Second, the results indicated that when *in agreement*, self- and meta-perceptions of Emotionality were negatively related to workplace well-being ($a_1$ = -0.13, $p$ = .021; see Fig 1g). Third, as self- and other-perceptions of Conscientiousness that were *in agreement* increased, workplace well-being *curvilinearly* increased ($a_2$ = 0.30, $p$ = .03; see Fig 1h).

## Post-hoc analysis

To our knowledge, the present study provided the first empirical evidence that self-perceptions of Honesty-Humility negatively predicted the degree of burnout symptoms. In order to conceptually grasp the relation between Honesty-Humility and burnout symptoms, we did a post-hoc analysis. As narrow traits of Honesty-Humility may have more explanatory strength [62], [see also 63], we examined correlations between the facets of Honesty-Humility and the subscales of burnout (i.e., exhaustion, mental distance, cognitive disorder, and emotional disorder). As shown in S5 Table, all facets of Honesty-Humility (i.e., sincerity, fairness, greed avoidance, and modesty) showed in particular negative correlations with the *mental distance* subscale of burnout (respectively $r$ = —.26, -.31, -.29, and -.25, $p$'s < .001).

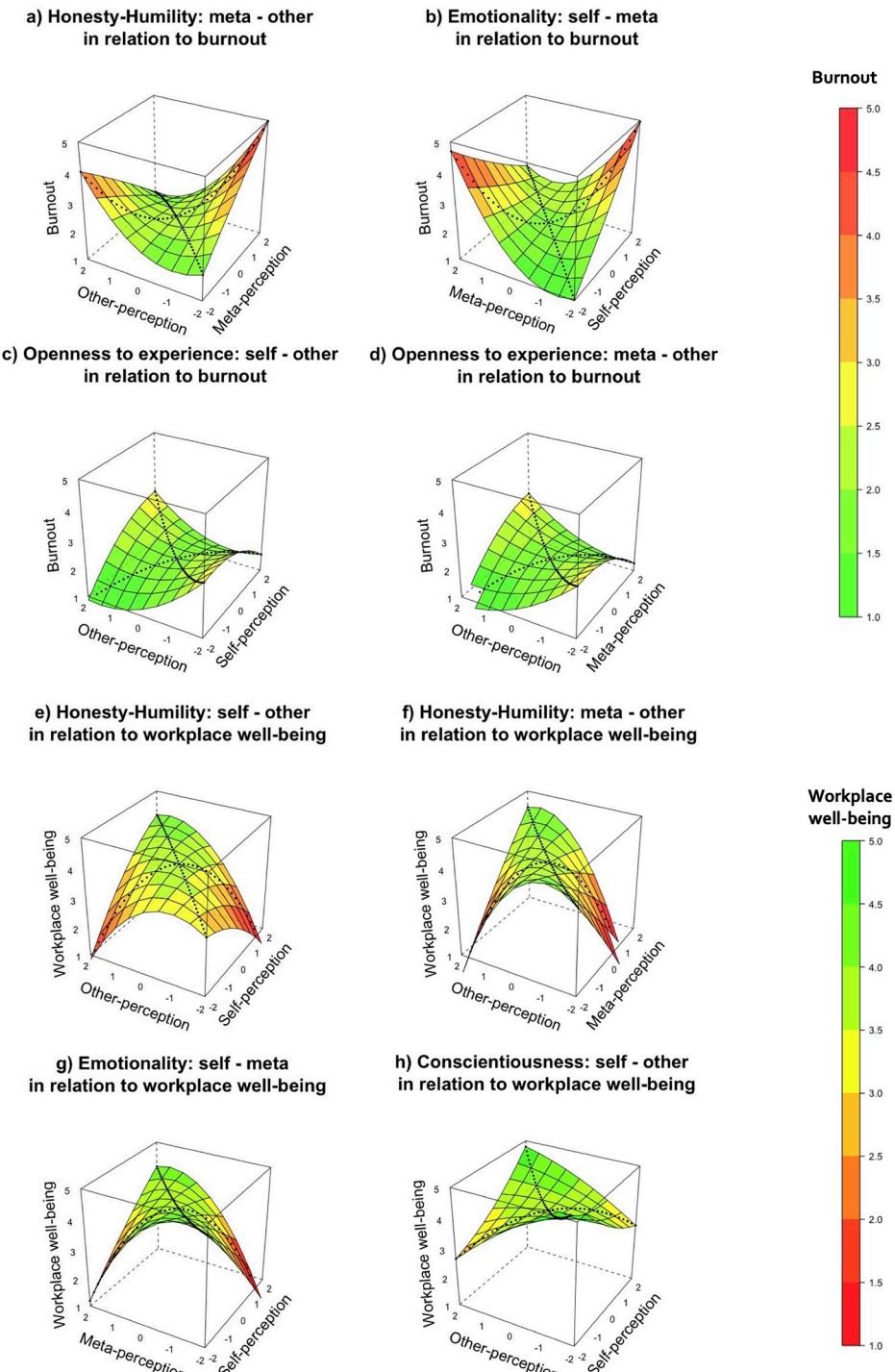

**Fig 1. Combinations of self-, other-, and meta-perceptions of personality related to burnout symptoms and eudaimonic workplace well-being.** Note: Dotted lines are line of agreement and line of incongruence.

**Table 3. Self-, other-, and meta-perceptions of HEXACO personality traits in relation to eudaimonic workplace well-being.**

| HEXACO factor scales | Perceptions $X{-}Y$ | Best-fitting model ($R^2$) | $X$ | $Y$ | $X^2$ | $XY$ | $Y^2$ |
|---|---|---|---|---|---|---|---|
| H | self—other | Model 3 (.07) | .09, $p = .553$ | .20, $p = .274$ | -.11, $p = .223$ | .37, $p = .019$ | -.31, $p = .022$ |
| | | | | $a_1 = 0.29, p = .224$ | $a_2 = -0.05, p = .756$ | $a_3 = -0.11, p = .652$ | $\boldsymbol{a_4 = -0.79, p = .005}$ |
| | meta—other | Model 3 (.09) | -.13, $p = .408$ | .25, $p = .153$ | -.04, $p = .698$ | .58, $p = .002$ | -.45, $p = .002$ |
| | | | | $a_1 = 0.12, p = .555$ | $a_2 = 0.09, p = .541$ | $a_3 = -0.38, p = .142$ | $\boldsymbol{a_4 = -1.07, p = .002}$ |
| | self—meta | Model 2 (.06) | .11, $p = .206$ | **.18, $p = .037$** | | | |
| | | | [-0.06, 0.27] | [0.01, 0.35] | | | |
| E | self—other | Model 2 (.05) | **-.13, $p = .031$** | -.12, $p = .190$ | | | |
| | | | [-0.26, 0.01] | [-0.29, 0.06] | | | |
| | meta—other | Model 2 (.03) | .06, $p = .410$ | **-.24, $p = .016$** | | | |
| | | | [-0.09, 0.21] | [-0.43, -0.04] | | | |
| | self—meta | Model 3 (.07) | -.18, $p = .064$ | .05, $p = .627$ | -.03, $p = .823$ | .46, $p = .107$ | -.33, $p = .058$ |
| | | | | $\boldsymbol{a_1 = -0.13, p = .021}$ | $a_2 = 0.10, p = .273$ | $a_3 = -0.23, p = .237$ | $a_4 = -0.82, p = .139$ |
| X | self—other | Model 2 (.17) | **.47, $p < .001$** | -.10, $p = .212$ | | | |
| | | | **[0.35, 0.61]** | [-0.26, 0.06] | | | |
| | meta—other | Model 2 (.10) | **.38, $p < .001$** | -.06, $p = .511$ | | | |
| | | | **[0.23, 0.53]** | [-0.23, 0.12] | | | |
| | self—meta | Model 2 (.15) | **.40, $p < .001$** | .02, $p = .839$ | | | |
| | | | **[0.22, 0.58]** | [-0.17, 0.21] | | | |
| A | self—other | Model 2 (.06) | **.24, $p < .001$** | -.09, $p = .174$ | | | |
| | | | **[0.12, 0.35]** | [-0.22, 0.04] | | | |
| | meta—other | Model 2 (.05) | **.22, $p < .001$** | -.11, $p = .113$ | | | |
| | | | **[0.10, 0.33]** | [-0.25, 0.03] | | | |
| | self—meta | Model 2 (.05) | **.17, $p = .033$** | .06, $p = .442$ | | | |
| | | | [0.01, 0.32] | [-0.09, 0.19] | | | |
| C | self—other | Model 3 (.07) | -.04, $p = .790$ | -.12, $p = .436$ | .07, $p = .568$ | .29, $p = .227$ | -.06, $p = .722$ |
| | | | | $a_1 = -0.15, p = .409$ | $\boldsymbol{a_2 = 0.30, p = .030}$ | $a_3 = 0.08, p = .704$ | $a_4 = -0.27, p = .550$ |
| | meta—other | Model 2 (.03) | .12, $p = .073$ | .02, $p = .779$ | | | |
| | | | [-0.01, 0.26] | [-0.15, 0.19] | | | |
| | self—meta | Model 2 (.05) | **.27, $p = .006$** | -.07, $p = .465$ | | | |
| | | | [0.08, 0.47] | [-0.24, 0.11] | | | |
| O | self—other | Model 2 (.03) | .11, $p = .052$ | -.04, $p = .613$ | | | |
| | | | [-0.00, 0.22] | [-0.18, 0.11] | | | |
| | meta—other | Model 2 (.03) | .18, $p = .010$ | -.08, $p = .330$ | | | |
| | | | [0.04, 0.31] | [-0.23, 0.08] | | | |
| | self—meta | Model 2 (.03) | .00, $p = .983$ | .13, $p = .156$ | | | |
| | | | [-0.16, 0.16] | [-0.05, 0.31] | | | |

*Note*: HEXACO factor scales are Honesty-Humility (H), Emotionality (E), Extraversion (X), Agreeableness (A), Conscientiousness (C), and Openness to Experience (O). $R^2$ refers to the variance explained of the model. The table represents the estimates and 95% confidence intervals of unstandardized regression coefficients, with significant effects ($p < .05$) in bold. Model 1 = control variables (i.e., age and gender). Model 2 = control variables + $X$ and $Y$. Model 3 = control variables + $X$, $Y$, $X^2$, $XY$, and $Y^2$. The model coefficients of Model 3 are not interpretable in isolation, but used to compute $a_1$ –$a_4$. Surface test values $a_1$ and $a_2$ represent the slope and curvature of the line of agreement and $a_3$ and $a_4$ represent the slope and curvature of the line of incongruence.

## Discussion

The main purpose of this study was to explore whether disagreement between self-, other-, and meta-perceptions of personality was related to burnout symptoms (an indicator of poor employee well-being; [15, 16]) and to well-being at work from a eudaimonic perspective, encompassing both intrapersonal and interpersonal dimensions of workplace well-being [17]. Our requirement was that disagreement in personality perceptions should explain incremental variance in burnout symptoms and eudaimonic workplace well-being that goes *beyond* the main effects of the different personality ratings. This was particularly relevant for our study given the large amount of evidence from meta-analyses that self-perceptions of personality traits are robust predictors of burnout [36, 55] and well-being [56]. Although we found mostly main effects of self-rated personality traits, this current study has also identified some effects of disagreement between perceptions of personality on burnout symptoms and eudaimonic workplace well-being.

### Disagreement between personality perceptions

In the present study, we expected that disagreement between personality perceptions, especially between meta- and other-perceptions and between self- and meta-perceptions, was *positively* related to burnout symptoms and *negatively* related to eudaimonic workplace well-being. Results showed little evidence on incremental effects of disagreement between personality perceptions, with one clear exception: when meta- and other-perceptions of Honesty-Humility became increasingly discrepant, burnout increased and eudaimonic workplace well-being decreased. Interestingly and (in contrast with the results for eudaimonic workplace well-being), the degree of discrepancy between self- and other-perceptions on Honesty-Humility was *not* associated with burnout symptoms. So, when colleagues rated the respondents' position on the Honesty-Humility dimension differently than the respondents rated themselves on this dimension, this did not lead to more distress by those respondents. Only when respondents *misjudged* how their colleagues would rate them on Honesty-Humility (i.e., discrepancy between meta- and other-perceptions), respondents experienced more feelings of burnout. This result demonstrates the added value of insight into how someone thinks he or she is viewed by others on Honesty-Humility (i.e., meta-perception) and the extent to which this corresponds to other-perceptions. The found effects of disagreement between meta- and other-perceptions of Honesty-Humility are in line with a body of literature suggesting that moral impressions of each other are the core of interpersonal perception and that moral impressions have meaningful (inter)personal consequences [see 64]. Especially, research by Barranti et al. [64] showed that individuals who were unaware of each other's judgements on moral character (e.g., honesty and loyalty) experienced more negative interpersonal outcomes. Overall, our results provide evidence that discrepancies between meta- and other-perceptions of Honesty-Humility affect employee well-being (i.e., burnout symptoms and eudaimonic workplace well-being).

### Main effects of self-rated personality traits on burnout symptoms and eudaimonic workplace well-being

Rather than effects of discrepancies between perceptions across the six personality traits, we found mostly main effects of *self-rated* personality traits on both burnout symptoms and eudaimonic workplace well-being. This was not unexpected, given the evidence from meta-analyses that self-perceptions of personality traits are strong predictors of burnout and well-being [36, 56, 55]. First, the present study has shown that lower levels of self-rated

Emotionality and higher levels of self-rated Honesty-Humility, Extraversion, Agreeableness, and Conscientiousness predicted lower burnout symptoms. With the exception of Honesty-Humility, these findings are consistent with previous research on the associations between Big Five personality traits and burnout [36, 55]. Yet, the current study provides an important extension of the literature, given that we used the six HEXACO personality dimensions to assess personality, demonstrating the utility of the added sixth personality dimension Honesty-Humility in better predicting and understanding employee burnout.

Although Honesty-Humility previously has been identified as a possible moderator of the relation between job demands and exhaustion [65, 66], this study provided, as far as we know, the first empirical evidence that Honesty-Humility negatively predicts burnout symptoms. Specifically, Honesty-Humility yielded relatively strong negative relations with the *mental distance* subscale of burnout. Mental distance in this context is expressed as a strong aversion to work. Employees withdraw mentally—or sometimes even physically—and avoid contact with others (e.g., colleagues). Characteristics of mental distancing are an indifferent and cynical attitude and no enthusiasm and interest for work [15]. One possible explanation for the negative relation between Honesty-Humility and burnout symptoms is that employees who score low on Honesty-Humility consider power and status to be important, flatter others to get what they want, show manipulative and unfair behaviors towards colleagues, and are inclined to break rules for personal profit [47]. This may result in less interest for work and a sense of cynicism about the workplace.

Second, much of the current research on well-being at work used generalized well-being measurements. Yet, we used a recently validated instrument to capture well-being *at work* from the *eudaimonic* perspective [17]. The current study shows that in particular self-ratings of Extraversion contribute uniquely to the prediction of eudaimonic workplace well-being. This finding is in line with results of recent meta-analytic research on the advantages (i.e., motivational, emotional, interpersonal, and performance advantages) of being extraverted at work [67]. Especially the enthusiasm and assertiveness aspects and the positive emotions, dominance, and activity aspects of Extraversion contribute to these advantages at work [67].

## Agreement between personality perceptions

Although the main focus of our study was to examine the effects of discrepancies between personality perceptions, polynomial regression analysis also allowed interpretation of how agreement between two personality perceptions (i.e., the levels of the two personality perceptions are essentially the same) relates to burnout symptoms and eudaimonic workplace well-being. First, when self- and meta-perceptions of Emotionality—that were in agreement—increased, burnout increased and eudaimonic workplace well-being decreased. Persons with high scores on Emotionality experience anxiety in response to stress, feel a need for emotional support from others, and feel sentimental attachments to others [47]. Conversely, persons with low scores on Emotionality feel little worry in stressful situations, have little need to share their concerns with others, and feel emotionally detached from others [47]. Not surprisingly, our study showed that self-rated Emotionality is positively related to burnout symptoms and negatively related to inter- and intrapersonal workplace well-being. In addition, when employees thought of themselves as highly emotional (i.e., self-perception) *and* they thought that colleagues also view them as highly emotional (i.e., meta-perception), this apparently resulted in more burnout symptoms and less well-being at work. This may imply that these employees are extra worried about the fact that colleagues see them as very emotional and that this may lead to more burnout symptoms and less eudaimonic workplace well-being.

Second, when in agreement, as the combined level of self- and other-perceptions with respect to Conscientiousness increased, eudaimonic workplace well-being *curvilinearly* increased. Congruent combinations of extreme (versus moderate) levels of Conscientiousness predicted higher levels of eudaimonic workplace well-being. So, when both employees and colleagues provided *high* scores on Conscientiousness, eudaimonic workplace well-being increased. Likewise, when both employees and colleagues provided *low* scores on Conscientiousness, eudaimonic workplace well-being also increased. In general, people prefer to be understood by others according to how they see themselves [3, 4]. Therefore, it could be possible that when an employee and his or her colleagues agree that the employee in question is *less* conscientious, this will lead to a sense of understanding and thereby enhance well-being. Agreement on being less conscientious could also imply experiencing lower work pressure (e.g., accepting that somebody will do less work). However, this curvilinearly found relation indicates that the relation between self- and other-perceptions of Conscientiousness on the one hand and eudaimonic workplace well-being on the other hand may be somewhat complex. Therefore, this exploratory finding should be interpreted with caution until replicated by future studies.

Third, congruent combinations of extreme levels of self- or meta-perceptions on the one hand and other-perceptions of Openness to Experiences on the other hand, predicted more burnout symptoms. So, when both employees and colleagues provided *high* scores on Openness to Experiences, burnout increased. Likewise, when both employees and colleagues provided *low* scores on Openness to Experiences, burnout also increased. Interestingly, meta-analytic research on the relation between personality and burnout [e.g., 36] found little evidence for Openness to Experiences as predictor of burnout. This might be due to the *curvilinear* relation between Openness to Experiences and burnout found in this study. Respondents very low on Openness to Experiences feel little intellectual curiosity, avoid creative pursuits, and feel little attraction toward ideas that may seem unconventional [47]. It may be possible that employees very low on Openness to Experiences experience feelings of distress if they are confronted with the demands of their work, especially in de case where employees correctly estimate that their colleagues also judge them low on Openness to Experiences. Conversely, respondents very high on Openness to Experience are inquisitive about various domains of knowledge, use their imagination freely in everyday life, and take an interest in unusual ideas or people [47]. Previous research showed that people who are too open to change are more likely to experience burnout [68, 69]. In sum, our findings suggest a curvilinear relation between Openness to Experiences and burnout symptoms in which agreement between self- and other-perceptions and between meta- and other-perceptions adds value in predicting burnout.

### Limitations and future directions

As with any study, we recognize the existence of certain limitations. First, contrary to our expectation that disagreement between self- and meta-perceptions was positively related to burnout symptoms and negatively related to eudaimonic workplace well-being, we did not find any evidence for this effect. It is possible that in the current study, when self-perceptions did not match meta-perceptions, the discrepancy between the two was not large enough to affect employee well-being. More research is needed to investigate the possible effects of large differences between self- and meta-perceptions, in- and outside the work context.

Second, in line with most studies using informant reports of personality [see 70], the informants (i.e., colleagues) in our study were selected by the participants themselves. The colleagues, who were selected by the participant, may tend to like the participant and thus might

portray him/her in specific ways (e.g., too positively). In this sense, the informant reports resembles more a *letter of recommendation* than an accurate, objective judgement of the employee's personality traits [71]. Previous research has shown that informants who liked their target more, evaluated them more positively (i.e., as being more extraverted, agreeable, open, conscientious, and less neurotic; [71]). In line with these findings, our study showed that respondents were structurally judged as more extraverted and more conscientious by their colleagues than respondents judged themselves and respondents estimated how colleagues would judge them. With respect to the other personality dimensions, we did not find evidence that colleagues tend to portray the respondents too positively. In contrast, colleagues generally judged the respondents lower on Honesty-Humility and Openness to Experience than respondents judged themselves. Nevertheless, when possible, the approach of selecting informants at random may be preferable instead of allowing participants to select informants of their own choice.

Third, another limitation of our study may be the use of same-source predictors and criteria, in such that we measured self-reported personality and both self-reported burnout symptoms and eudaimonic workplace well-being. This leaves our results vulnerable to common-method variance [e.g., 72]. Although many researchers assume that common-method variance is a serious problem in organizational research, others have questioned whether this assumption is correct [for details, see 73]. In reference to the current research, self-reports might in fact be the most valid measurement method, because a participant is the best person to report on his/her own level of burnout symptoms and eudaimonic workplace well-being. Yet, future research may use a mix of self-reported and other-reported criteria, such as how (other) colleagues experience the collaboration with the participant. It is also possible that (dis)agreement on the personality traits is less relevant for employees' degree of burnout and eudaimonic workplace well-being, but more relevant to other work-related outcomes, such as interpersonal citizenship behavior, "getting along" performance, self-efficacy and managerial effectiveness. Thus, more research is needed to better understand the effects of disagreement between personality perceptions in a work environment.

Fourth, we did not examine potential moderators that may shape the effects of disagreement between self-, other-, and meta-perceptions, such as organization culture, managerial versus non-managerial positions, and the quality of the relationship between respondents and their colleagues. Also, the results do not answer the question of causality: do discrepancies between personality perceptions of Honesty-Humility precede burnout symptoms and eudaimonic workplace well-being or vice versa? More longitudinal research is necessary to answer the question of causality.

## Methodological strengths and further directions

This study has several methodological strengths. First, previous studies gathering personality ratings from informants often used only one informant, instead of two or more [see 70]. This is problematic since using one informant report in order to measure other-perceptions is not likely to be reliable in psychometric sense [70]. Our current study has illustrated the increase in reliability using multiple informants instead of only one. We therefore recommend future researchers to obtain other-perceptions of multiple informants and to use the Koo and Li [74] guideline to calculate the intraclass correlation coefficient.

Second, previous research on disagreement in personality perceptions typically relied on methods that used profile similarity indices or absolute difference scores. However, these commonly used methods may produce incomplete and/or inaccurate results [see 52]. We used an alternative methodology: polynomial regression analysis with response surface analysis [53],

[see also 54]. This approach allows researchers to examine the extent to which *combinations* of two predictor variables relate to an outcome variable, particularly in the case when the discrepancy between two predictor variables is a central consideration [54]. In order to reduce collinearity in polynomial regression analysis, researchers frequently center the variables around the mean, before creating the interaction and nonlinear terms [75]. However, when mean-centering is used, it substantially complicates interpretation [60], [see also 75]. Therefore, centering on the *midpoint of the scale* is recommended for polynomial regression analysis with response surface analysis [52], which ensures accurate interpretation of the results of how (dis)agreement between two variables relates to the outcome variable [60]. So, future researchers should be careful that they do not center predictors on the scale mean, because it may change the interpretation of the response surface analysis.

## Practical implications

From a practical perspective, our findings are in line with meta-analyses that underline the role of personality factors as individual predictors of burnout [36, 55]. Organizations could use personality assessment to identify employees who are prone to burnout (i.e., employees who score low on Extraversion and Conscientiousness, and high on Emotionality) and offer them some form of stress reduction training. In the case of a low score on Honesty-Humility, organizations should be especially aware that these employees could be cynical about work and may be less interested in their work. Considering that Honesty–Humility has been found to be negatively related to all kinds of counterproductive or delinquent behaviors at work [e.g., 50, 51], employers may be even advised to select on Honesty–Humility to protect the organization and its employees. Even though selection should not be based solely on applicant questionnaire scores on Honesty-Humility, it can be taken into account since it seems to be a protective factor. The Honesty-Humility scores and its potential protective function could for example be discussed with applicants in a post questionnaire interview.

As the findings of our analyses provided some evidence for effects of disagreement and agreement between personality perceptions, it might be insightful for employees and their colleagues, e.g. in team development workshops, to share their inside (how do I see myself?) and outside (how do I view you?) personality perspectives and their meta-perceptions (how do I think I am viewed by my colleagues?). In this way, employees can learn more about themselves, about how they are perceived by colleagues, and about their (in)accurate assumptions about how they are seen at work. This reflection may help employees to gain more self-knowledge, which can help them to make behavioral changes in response to social cues at work, in order to reduce sources of stress and to improve mental health. It also may help employees to understand each other better.

## Conclusion

To sum up, the main purpose of this study was to explore whether disagreement between self-, other-, and meta-perceptions of the six HEXACO personality dimensions was positively related to burnout symptoms and negatively related to eudaimonic workplace well-being. The results, based on polynomial regression analyses with response surface analyses, highlighted strong main effects of self-rated personality traits in relation to burnout symptoms and eudaimonic workplace well-being. This study provided, as far as we know, the first empirical evidence that self-ratings of Honesty-Humility negatively predict burnout symptoms. Results showed little evidence on incremental effects of disagreement between personality perceptions, with one clear exception: when meta- and other-perceptions of Honesty-Humility became increasingly discrepant, burnout increased and eudaimonic workplace well-being decreased.

Thus, discrepancies between meta- and other-perceptions on Honesty-Humility affect employee well-being. This finding demonstrates the added value of insight into how someone thinks he or she is viewed by others on Honesty-Humility and the extent to which this corresponds to other-perceptions.

## Supporting information

**S1 Text. Data checks.**
(DOCX)

**S1 Table. Results Intraclass Correlation Coefficients (ICC).**
(TIF)

**S2 Table. Correlations, Cronbach's alpha reliabilities (on diagonal), and descriptives of variables.**
(TIF)

**S3 Table. Results of facet-level principal components analysis: Meta-perception.**
(TIF)

**S4 Table. Frequencies agreement groups.**
(TIF)

**S5 Table. Correlations Honesty-Humility facets and burnout subscales.**
(TIF)

**S1 Data.**
(SAV)

## Acknowledgments

The authors would like to acknowledge Ariëtta Groeneveld, Daniëlle Driest, Esther Renes, Femke Stoltenborg, Frank Huisman, Jan Bolle, Kim Oomen, Liselot Koning, Lisette Christenhuis, Mark van Dijk, Merel van Dongen, Roselein de Graaf, Shivan P.S. Nandoe, Tanja Vons, and Wanda Bodewitz for their help in data collection. The authors also thank Jolanda de Vries for her great help in merging the data files and Wim K. B. Hofstee (†) for inspiration.

## Author Contributions

**Conceptualization:** Anita de Vries, Wim Bloemers, Jeroen Kuntze.

**Data curation:** Anita de Vries, Jeroen Kuntze.

**Formal analysis:** Anita de Vries, Vera M. A. Broks.

**Investigation:** Anita de Vries.

**Methodology:** Anita de Vries, Vera M. A. Broks.

**Project administration:** Anita de Vries.

**Software:** Vera M. A. Broks.

**Supervision:** Anita de Vries, Wim Bloemers, Reinout E. de Vries.

**Visualization:** Vera M. A. Broks.

**Writing – original draft:** Anita de Vries, Wim Bloemers, Jeroen Kuntze, Reinout E. de Vries.

**Writing – review & editing:** Anita de Vries, Vera M. A. Broks, Wim Bloemers.

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
