## [Decision Letter · Decision Letter 0]

27 May 2022

PONE-D-21-35155Self-, other-, and meta-perceptions of personality: relations with burnout and eudaimonic workplace well-beingPLOS ONE

Dear Dr. Bloemers,

Thank you for submitting your manuscript to PLOS ONE. After careful consideration, we feel that it has merit but does not fully meet PLOS ONE’s publication criteria as it currently stands. Therefore, we invite you to submit a revised version of the manuscript that addresses the points raised during the review process. Please note that we have only been able to secure a single reviewer to assess your manuscript. We are issuing a decision on your manuscript at this point to prevent further delays in the evaluation of your manuscript. Please be aware that the editor who handles your revised manuscript might find it necessary to invite additional reviewers to assess this work once the revised manuscript is submitted. However, we will aim to proceed on the basis of this single review if possible. 

Additionally, please note that in addition to the reviewer comments below, based on further correspondence with the reviewer we also ask you to address the following: 1) S2 Table: please add a top row with the variable number for easier reference. 2) S5 Table: please also include the overall burnout score in line with BAT's presentation that burnout should also be considered an overall score

We look forward to receiving your revised manuscript.

Kind regards,

Hugh Cowley

Senior Editor

PLOS ONE

Journal Requirements:

4. We note that you have referenced (Schaufeli WB, De Witte et al. [15]) which has currently not yet been accepted for publication. Please remove this from your References and amend this to state in the body of your manuscript: (Schaufeli WB, De Witte et al. [15 [Unpublished]”) as detailed online in our guide for authors

Reviewers' comments:

Reviewer's Responses to Questions

**Comments to the Author**

1. Is the manuscript technically sound, and do the data support the conclusions?

Reviewer #1: Yes

2. Has the statistical analysis been performed appropriately and rigorously? 

Reviewer #1: I Don't Know

3. Have the authors made all data underlying the findings in their manuscript fully available?

Reviewer #1: No

4. Is the manuscript presented in an intelligible fashion and written in standard English?

Reviewer #1: Yes

5. Review Comments to the Author

Reviewer #1: This article was very interesting, utilizing the new conceptualization of burnout - and also the associated measurement the BAT-23.

1) Can you not be more specific in the abstract, given the results inside the article itself:

"Only when respondents misjudged how their colleagues would rate them on Honesty-Humility (i.e., discrepancy between meta- and other-perceptions), respondents experienced more

feelings of burnout."

"Overall, our results provide evidence that discrepancies between meta- and other-perceptions of Honesty-Humility affect employee well-being (i.e., burnout and eudaimonic workplace well-being)."

2) Is the sentence in 505-508 correct? It reads strange to me. I assume the "this" should be "his". Furthermore, I am not sure if this "understanding" explanation here is necessarily accurate. Could it not be that because less conscientious that they do not feel as much pressure in the workplace? Just a thought.

3) Line 606. I think the caveat should just be added that selecting only on one trait should of course be done with caution, but that it does seem to be a protective factor according to your results.

4) Lines 608+ is a very interesting and practical suggestion. I know some consultants present team development workshops for organizations that help employees understand each other better. Perhaps you can state it (somewhat) as blatantly as that.

5) Is there any research on honesty-humility and depression? The reason I am asking this is because of the criticism that burnout as a concept is receiving (that it is just depression) if your results don't perhaps reveal something interesting compared to past findings. If there is something a short addition at the appropriate instance in the manuscript might be useful. This is just a strong suggestion, I don't want to force it on you. But if there is something, as I said, this would be interesting in the context of your results.

6) To me it is not entirely clear how you constructed your burnout score. I think it is important to state how you constructed it from its 4 components. I was also unable to see the supplementary material only the main manuscript.

7) Be sure to submit high quality images for the Figures that will show well in a PDF document. I found it hard to read some of them.

8) "alpha reliabilities" add Cronbach's before alpha.

6. PLOS authors have the option to publish the peer review history of their article (what does this mean?). If published, this will include your full peer review and any attached files.

Reviewer #1: No

---

## [Author Response · Author response to Decision Letter 0]

28 Jun 2022

We thanks the editor and the reviewer for their insightful and helpful comments. As far as we can see, we have complied with all suggestions and remarks, thus improving the quality of our manuscript. We are looking forward to the next response. 

Yours sincerely, Wim Bloemers, corresponding author.

---

## [Decision Letter · Decision Letter 1]

13 Jul 2022

Self-, other-, and meta-perceptions of personality: relations with burnout and eudaimonic workplace well-being

PONE-D-21-35155R1

Dear Dr. Bloemers,

We’re pleased to inform you that your manuscript has been judged scientifically suitable for publication and will be formally accepted for publication once it meets all outstanding technical requirements.

Kind regards,

Norio Yasui-Furukori

Academic Editor

PLOS ONE

Additional Editor Comments (optional):

Thank you for submitting the revised manuscript. I have verified that it has been properly corrected. I have no additional comments.

Reviewers' comments:

Reviewer's Responses to Questions

**Comments to the Author**

1. If the authors have adequately addressed your comments raised in a previous round of review and you feel that this manuscript is now acceptable for publication, you may indicate that here to bypass the “Comments to the Author” section, enter your conflict of interest statement in the “Confidential to Editor” section, and submit your "Accept" recommendation.

Reviewer #1: All comments have been addressed

2. Is the manuscript technically sound, and do the data support the conclusions?

Reviewer #1: Yes

3. Has the statistical analysis been performed appropriately and rigorously? 

Reviewer #1: Yes

4. Have the authors made all data underlying the findings in their manuscript fully available?

Reviewer #1: Yes

5. Is the manuscript presented in an intelligible fashion and written in standard English?

Reviewer #1: Yes

6. Review Comments to the Author

Reviewer #1: Thank you for the opportunity to view this manuscript again. My comments have been addressed sufficiently by the authors.

7. PLOS authors have the option to publish the peer review history of their article (what does this mean?). If published, this will include your full peer review and any attached files.

Reviewer #1: No

---

## [Editor Report · Acceptance letter]

19 Jul 2022

PONE-D-21-35155R1 

Self-, other-, and meta-perceptions of personality: relations with burnout symptoms and eudaimonic workplace well-being 

Dear Dr. Bloemers:

I'm pleased to inform you that your manuscript has been deemed suitable for publication in PLOS ONE. Congratulations! Your manuscript is now with our production department. 

Kind regards, 

on behalf of

Dr. Norio Yasui-Furukori 

Academic Editor

PLOS ONE